

# Expression of thermophilic two-domain laccase from *Catenuloplanes japonicus* in *Escherichia coli* and its activity against triarylmethane and azo dyes

Liubov Igorevna Trubitsina[1], Azat Vadimovich Abdullatypov[2],
Anna Petrovna Larionova[1,3], Ivan Vasilyevich Trubitsin[1],
Sergey Valerievich Alferov[4], Olga Nikolaevna Ponamoreva[4] and
Alexey Arkadyevich Leontievsky[1,3]

[1] G. K. Skryabin Institute of Biochemistry and Physiology of Microorganisms, Russian Academy of Sciences – A Separate Subdivision of PSCBR RAS (IBPM RAS), Pushchino, Moscow Region, Russian Federation
[2] Institute of Basic Biological Problems of the Russian Academy of Sciences – A Separate Subdivision of PSCBR RAS (IBBP RAS), Pushchino, Moscow Region, Russian Federation
[3] Pushchino State Institute of Natural Sciences, Pushchino, Moscow Region, Russian Federation
[4] Department of Biotechnology, Tula State University, Tula, Tula Region, Russian Federation

Corresponding author
Liubov Igorevna Trubitsina,
lyubov_yurevich@mail.ru

## ABSTRACT

**Background:** Two-domain laccases are copper-containing oxidases found in bacteria in the beginning of 2000ths. Two-domain laccases are known for their thermal stability, wide substrate specificity and, the most important of all, their resistance to so-called «strong inhibitors» of classical fungal laccases (azides, fluorides). Low redox potential was found to be specific for all the two-domain laccases, due to which these enzymes lost the researchers' interest as potentially applicable for various biotechnological purposes, such as bioremediation. Searching, obtaining and studying the properties of novel two-domain laccases will help to obtain an enzyme with high redox-potential allowing its practical application.

**Methods:** A gene encoding two-domain laccase was identified in *Catenuloplanes japonicus* genome, cloned and expressed in an *Echerichia coli* strain. The protein was purified to homogeneity by immobilized metal ion affinity chromatography. Its molecular properties were studied using electrophoresis in native and denaturing conditions. Physico-chemical properties, kinetic characteristics, substrate specificity and decolorization ability of laccase towards triphenylmethane dyes were measured spectrophotometrically.

**Results:** A novel two-domain recombinant laccase CjSL appeared to be a multimer with a subunit molecular mass of 37 kDa. It oxidized a wide range of phenolic substrates (ferulic acid, caffeic acid, hydroquinone, catechol, etc.) at alkaline pH, while oxidizing of non phenolic substrates ($K_4[Fe(CN)_6]$, ABTS) was optimal at acidic pH. The UV-visible absorption spectrum of the purified enzyme was specific for all two-domain laccases with peak of absorption at 600 nm and shoulder at 340 nm. The pH optima of CjSL for oxidation of ABTS and 2, 6-DMP substrates were 3.6 and 9.2 respectively. The temperature optimum was 70 °C. The enzyme was most stable in neutral-alkaline conditions. CjSL retained 53% activity after pre-incubation at 90 °C for 60 min. The enzyme retained 26% activity even after 60

min of boiling. The effects of NaF, NaN$_3$, NaCl, EDTA and 1,10-phenanthroline on enzymatic activity were investigated. Only 1,10-phenanthroline reduced laccase activity under both acidic and alkaline conditions. Laccase was able to decolorize triphenylmethane dyes and azo-dyes. ABTS and syringaldehyde were effective mediators for decolorization. The efficacy of dye decolorization depended on pH of the reaction medium.

## INTRODUCTION

Laccase (benzenediol:oxygen oxidoreductase, EC 1.10.3.2) belongs to the family of multicopper oxidases. It contains four copper ions in the active site. A total of four copper atoms assembled into three metal centers (mononuclear T1 and T2, binuclear T3) take part in the catalytic act (*Solomon, Sundaram & Machonkin, 1996*; *Bento et al., 2005*). A total of four electrons required for complete reduction of oxygen to water are subsequently taken away from the substrates (a wide range of organic and inorganic compounds) (*Morozova et al., 2007*). Due to wide substrate range, high stability and high redox-potential laccases are widely used in different biotechnological processes, such as detoxification of organic pollutants (*Torres, Bustos-Jaimes & Le Borgne, 2003*), delignification (*Leonowicz et al., 2001*), pulp bleaching (*Ibarra et al., 2006*), decolorization of dyes (*Wong & Yu, 1999*). Laccases are also used in the food industry, in organic synthesis, in cosmetics and in medicine (*Rodríguez Couto & Toca Herrera, 2006*).

Laccases were found in fungi (*Baldrian, 2006*), plants (*Joel, Marbach & Mayer, 1978*; *Gavnholt & Larsen, 2002*), insects (*Dittmer & Kanost, 2010*), lichens (*Lisov et al., 2007*), algae (*Otto & Schlosser, 2014*), and bacteria (*Sharma, Goel & Capalash, 2006*). Bacterial laccases can contain two or three structural and functional domains. Contrary to the three-domain laccases, which are active in monomeric form, two-domain laccases are active in multimeric forms (homotrimers, homohexamers) (*Lisov et al., 2019*). All the 12 two-domain laccases characterized to date were found in bacteria belonging to the phylum Actinobacteria, the majority of them were isolated from *Streptomyces*. Deletion of one domain confers bacterial laccases unusual properties which can be used in biotechnological applications. Two-domain laccases are thermostable (*Feng et al., 2015*; *Trubitsina et al., 2015*), resistant to alkaline pH values (*Gunne & Urlacher, 2012*; *Feng et al., 2015*; *Lisov et al., 2019*), tolerant to high concentrations of NaCl and to typical laccase inhibitor sodium azide (*Molina-Guijarro et al., 2009*; *Feng et al., 2015*; *Trubitsina et al., 2015*). They are used (in complex with mediators) in detoxification of dyes (*Dube et al., 2008*; *Molina-Guijarro et al., 2009*; *Blánquez et al., 2019*). The disadvantageous property of the studied two-domain laccases is their low redox potential (0.35–0.45 V) significantly restricting the use of these enzymes in practical applications. Taking into account the unusual physico-chemical properties as well as a small number of characterized members of this group, searching, obtaining and investigating new two-domain proteins with the

desired properties is a promising task. Availability of nucleotide sequences of two-domain laccases in databases allows rapid cloning of the desired protein and its further characterization.

The aims of this work were cloning, expression and characterization of the two-domain laccase from *Catenuloplanes japonicus* VKM Ac-875, investigation of its physico-chemical properties and substrate specificity, determination of the enzyme's redox potential and estimation of applicability of this laccase in bioremediation (investigation of the enzyme's ability to decolorize various dyes).

## MATERIALS & METHODS

### Microorganism, and cloning of cjsl gene

Strain *Catenuloplanes japonicus* VKM Ac-875 was obtained from the All-Russian collection of microorganisms (http://www.vkm.ru/Collections.htm). The strain was grown on peptone yeast agar media, and its genomic DNA was purified from the biomass using diaGene kit for genomic DNA isolation from bacterial cell cultures (Dia-M, Moscow, Russia). Primers for PCR were designed based on the predicted multicopper oxidase sequence from the genome of *Catenuloplanes japonicus* NRRL B-16061 (NCBI Reference Sequence of protein: WP_033344226.1). Using the designed primers (875F 5′-ATGGA CGACAACGTTGACAAACC-3′ and 875R 5′-TCATCCGGTGTGCCCTCC-3′), a corresponding PCR product was amplified. Correct amplification of the target gene was verified by sequencing. To clone the gene into expression vector, primers 875Fe (5′-AGTGGATCCGCGGGAGCCACCCGGAAG-3′) and 875Re (5′-AGTAAGCTTTCA TCCGGTGTGCCCTCC-3′) were designed (cleavage sites for restriction endonucleases are underscored). The first primer introduced a BamHI site, and the second one inserted a HindIII site after the stop codon. The DNA fragment encoding *cjsl*, without the region for the signal sequence, was generated by PCR with 875Fe and 875Re primers and genomic DNA as template. The BamHI/HindIII-digested amplicon was cloned into pQE-30 (Qiagen, Hilden, Germany). Plasmid pQE::cjsl was used to transform *E. coli* M15 (pREP4) expression strain. Transformants were selected on LB plates containing 100 μg ampicillin/ml and 25 μg kanamycin/ml.

### Recombinant expression and CjSL purification

For the production of CjSL, strain *E. coli* M15 (pREP4) transformed with pQE::cjsl plasmid was grown at 37 °C with agitation at 250 rpm to a cell density of 0.2–0.25 ($A_{600}$). Then, 0.2 mM isopropyl-β-D-thiogalactopyranoside and 1 mM $CuSO_4$ were added to culture medium. After that, the cells were first incubated for 18 h at 20 °C with agitation at 100 rpm, then they were incubated for 24 h at 25 °C without shaking. Cells were collected by centrifugation at 4,000 g for 30 min, suspended in 20 ml of 20 mM Tris-HCl buffer, pH 8.0, containing 0.5 M NaCl and 1 mM imidazole (buffer A) and disrupted by sonication. Cell debris was removed by centrifugation (40 min at 8.000 g). The protein was purified by affinity chromatography on a HisTrap 5-ml column (GE Healthcare, Chicago, IL, USA). Cell extract was loaded onto a HisTrap column equilibrated with buffer A. After loading, the column was washed with four volumes of the buffer A and then washed
with four volumes of the buffer B (20 mM Tris-HCl, 0.5 M NaCl, 50 mM imidazole, pH 8.0). Active fractions containing the enzyme were eluted with buffer C (20 mM Tris-HCl, 0.5 M NaCl, 300 mM imidazole, pH 8.0). After the chromatography stage, the protein was dialyzed against 20 mM Tris-HCl buffer (pH 7.5) with 0.1 M NaCl, and then against 20 mM Tris-HCl buffer pH 7.5 without NaCl.

### Enzyme characterization

The concentration of the protein was determined using the molar extinction at 280 nm ($\varepsilon$ = 43,930 $M^{-1} \times cm^{-1}$) calculated from the protein sequence using Vector NTI Program (Life Technologies, Carlsbad, CA, USA). Presence of cupredoxin domains in the enzyme structure was identified using BLAST (http://blast.ncbi.nlm.nih.gov/Blast.cgi) and InterProScan (https://www.ebi.ac.uk/interpro/) services. Presence of signal peptide was also identified in InterProScan. The UV-Vis absorption spectrum and molecular weight of native and denatured protein were determined as described earlier (*Trubitsina et al., 2015*).

Laccase activity was determined at room temperature by measuring the oxidation of 1 mM 2,2'-azinobis(3-ethylbenzothiazoline-6-sulfonate) (ABTS) and 1 mM 2,6-dimethoxyphenol (2,6-DMP) in 50 mM Britton–Robinson buffer (*Britton & Robinson, 1931*) (at pH 3.6 and pH 9.2, respectively). The 50 mM Britton–Robinson buffer was prepared by mixing equal amounts of 0.05 M boric acid, 0.05 M orthophosphoric acid and 0.05 M acetic acid, and adjusting pH to the required value using 1 M NaOH. The oxidation of the substrates was detected by measuring the absorbance at 420 nm for ABTS ($\varepsilon$ = 36,000 $M^{-1} \times cm^{-1}$) (*Heinfling et al., 1998*) and at 469 nm for 2,6-DMP ($\varepsilon$ = 49,600 $M^{-1} \times cm^{-1}$) (*Wariishi, Valli & Gold, 1992*)

Optimal pH value was determined with ABTS (in a pH range from 3 to 5.5) and 2,6-DMP (in pH range from 8.0 to 10.5) in 50 mM Britton-Robinson buffer. The temperature dependence of the activity was determined in 50 mM Britton-Robinson buffer (pH 3.6) at temperatures from 30 °C to 90 °C using 1 mM 2,6-DMP as substrate. pH stability was estimated by incubation of the enzyme at room temperature for 15 days in the same buffer at pH values 3, 5, 7, 9, 11. Residual activity assay was determined in 50 mM Britton-Robinson buffer (pH 3.6) using 1 mM ABTS as substrate. Thermal stability of the enzyme was measured in 50 mM Britton-Robinson buffer (pH 3.6) using 1 mM ABTS at 80 °C, 90 °C and at water boiling temperature, by incubating the enzyme in thin-wall microtubes for 1 h (protein aliquots were taken every 10 min to determine the activity loss during the experiment).

The substrate specificity of the recombinant laccase was examined against 26 different putative substrates by detecting the changes in their absorption spectra after 24 h incubation. The substrate specificity was assayed at pH 3.6 or pH 9.2 for non-phenolic and phenolic compounds, respectively, in 50 mM Britton-Robinson buffer. The compounds assayed were: tyrosine, gallic acid (3,4,5-trihydroxybenzoic acid), ferulic acid (3-methoxy-4-hydroxycinnamic acid), vanillic acid (4-hydroxy-3-methoxybenzoic acid), tannic acid, caffeic acid (3,4-dihydroxycinnamic acid), gentisic acid (2,5-dihydroxybenzoic acid), syringic acid (4-hydroxy-3,5-dimethoxybenzoic acid), o-coumaric acid, 2-thiobarbituric

acid, 3,4-dihydroxybenzoic acid, 4-hydroxybenzoic acid, $K_4[Fe(CN)_6]$, ABTS, hydroquinone, methylhydroquinone, syringaldehyde (4-hydroxy-3,5-dimethoxybenzaldehyde), pyrogallol (01,2,3-trihydroxybenzene), guaiacol (2-methoxyphenol), catechol (1,2-dihydroxybenzene), vanillin (4-hydroxy-3-methoxybenzaldehyde), syringaldazine, 2,6-dimethoxyphenol, 3,5-dimethoxyphenol, 3,4,5-trimethoxyphenol, 2-aminophenol. The substrate concentration used was 1 mM.

Sodium azide ($NaN_3$), sodium fluoride (NaF), sodium chloride (NaCl), ethylenediaminetetraacetic acid (EDTA) and 1,10-phenanthroline were used as inhibitors. Inhibitor concentrations were 1, 10 and 100 mM in the reaction mixture with 1 mM ABTS or 1 mM 2,6-DMP as substrates in 50 mM Britton–Robinson buffer (pH 3.6 or 9.2 respectively).

The steady-state kinetic constants were obtained for the substrates 2,6-DMP and ABTS at the substrates' pH optima at 30 °C. Calculation of the apparent kinetic constants was performed by a nonlinear regression of the data using Sigma Plot 11.0 software.

## Dye decolorization

The decolorization of the following triarylmethane dyes: Malachite Green ($\lambda_{max}$ = 617 nm (pH 4.0)), Brilliant Green ($\lambda_{max}$ = 623 nm (pH 4.0)); azo dyes: Methyl Orange ($\lambda_{max}$ = 478 nm (pH 4.0); $\lambda_{max}$ = 464 nm (pH 6.5; 9.2)), Methyl Red ($\lambda_{max}$ = 524 nm (pH 4.0; 6.5); $\lambda_{max}$ = 432 nm (pH 9.2)), Congo Red ($\lambda_{max}$ = 570 nm (pH 4.0); $\lambda_{max}$ = 488 nm (pH 6.5; 9.2)), by laccase was tested. The reaction mixture (1 ml) contained 50 mM Britton–Robinson buffer (pH 4.0, 6.5 or pH 9.2), dye (final concentration of 50 μM) and purified enzyme 200 mU. A total of 50 μM ABTS and 50 μM syringaldehyde (SA) were used as potential redox mediators. Reactions mixture was incubated at 30 °C for 24 h. Control samples without enzyme were run in parallel under the same conditions. The dye degradation was judged by change of absorption spectrum of the oxidized compound and expressed in decolorization rate. The calculation formula: D = (A0−A1)/A0 × 100%, where D represents the decolorization rate (%/day), A0 is the initial absorbance of the dye solution at the maximum absorption wavelength, A1 results from the initial absorbance of the dye solution at the maximum absorption wavelength after the reaction.

## Homology modeling

An initial model of Ac-875 laccase was created using SwissModel automatic protein model builder (*Waterhouse et al., 2018*). A large set of models of Ac-875 laccase was built in MODELLER (*Webb & Sali, 2016*). An experimentally determined structure of another trimeric two-domain laccase from *Streptomyces coelicolor* (PDB ID: 3CG8) was used as a template for model building. Parts of the protein that were not included into the alignment were not included into the models. A trimeric alignment was used to generate trimeric models. The best models were sorted according to their z-DOPE score (result of assess_normalized_dope command) (*Shen & Sali, 2006*) taken on average for three subunit models. Model refinement was achieved by internal resources of MODELLER package, including simulated annealing, steepest descent and in vacuo molecular dynamics.

## Statistical analysis

Mean values and standard deviations were calculated for at least three replicates.

## RESULTS

### Protein cloning, expression and purification

Two-domain laccase gene was identified in the genome of actinobacterium *Catenuloplanes japonicus* NRRLB-16061 using BLAST program. Analysis of amino acid sequence in InterPro Scan revealed two cupredoxin domains (74–191, 195–307) and a signal peptide facilitating enzyme translocation via TAT-pathway (1–45). Conserved copper-binding residues, ten histidines and one cysteine, were all present in the protein sequence. The structural laccase gene designated as *cjsl* was cloned into pQE-30 expression plasmid without the TAT-leader sequence. pQE::cjsl construct was used for transformation of competent cells of *E. coli* M15 (pREP4). Laccase was expressed as a mature enzyme, without a signal peptide, with a 6 × His-tag at the N-terminus instead. Thankfully to one-step purification on Ni-sepharose column, laccase was obtained in electrophoretically homogenous state. Presence of copper in the medium during heterologous expression and a step of microaerobic cultivation of the strain for saturation of the culture by copper ions (*Durao et al., 2008*) were mandatory conditions for obtaining blue active enzyme. The yield of the protein was low, about 10 mg from one liter of culture.

### Enzyme characterization

Based on amino acid sequence of CjSL, the calculated molecular weight of the protein was 33.8 kDa. CjSL boiled with β-mercaptoethanol and SDS migrated at 37 kDa in SDS-PAGE, CjSL with β-mercaptoethanol and SDS without boiling migrated 70 kDa (Fig. 1A). Gradient native gel electrophoresis showed that the molecular weight of the native active protein was around 200 kDa (Fig. 1B). All this points on multimeric state of the native laccase. The absorption spectrum of the enzyme had a maximum at 600 nm due to the presence of T1-copper center conferring deep blue color, and the shoulder at 340 nm can be attributed to the presence T3-copper center (Fig. 2).

The enzyme catalyzed oxidation of an electron donor ABTS at acidic pH values. Maximal ABTS oxidation rate was observed at pH 3.6. Oxidation of an electron and proton donor phenolic compound 2,6-DMP by CjSL was observed at alkaline pH values with an optimum of pH 9.2 (Fig. 3A). The optimum reaction temperature for the recombinant laccase was 70 °C (Fig. 3B). In terms of thermal denaturation, CjSL turned out to be a very stable enzyme. After an hour of incubation at 80 °C, the laccase preserved 63% of its initial activity, at 90 °C, 53% of the initial activity, and 26% of the initial activity was conserved after an hour of boiling (Fig. 4). CjSL was most stable at pH 5 and 7. It retained around 67% and 60% of activity after 15 days of incubation at pH 5.0 and 7.0, respectively. At more acidic conditions, it was less stable. At pH 3.0, the residual activity of the enzyme by the end of 24 h of incubation was around 12% (Fig. 5).

The effect of some conventional laccase inhibitors (NaN₃, NaF, NaCl, EDTA and 1,10-phenanthroline) was studied. Table 1 summarizes the results. Inhibition effect was measured at acidic and alkaline conditions. It was established that only 1,10-

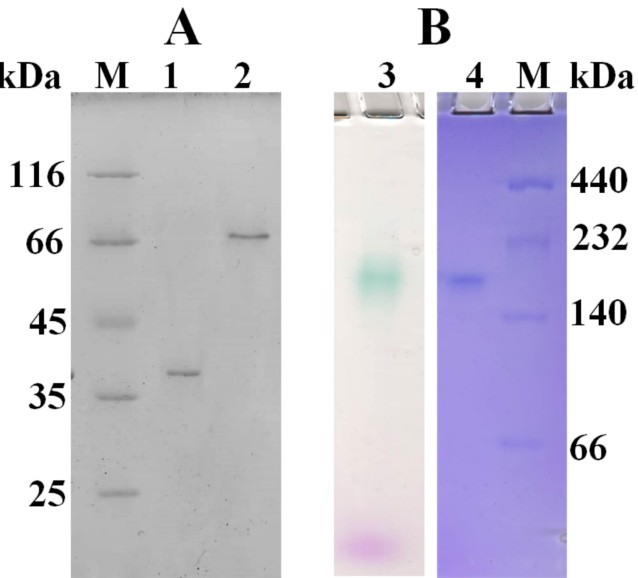

**Figure 1 SDS-PAGE (A) and native 3–15% gradient PAGE (B) of laccase from *C. japonicas*.** (M)––molecular weight markers; (1)–purified enzyme boiled with β-mercaptoethanol and SDS; (2)–enzyme with β-mercaptoethanol and SDS but without boiling; (3)–zymogram of laccase activity with 2.6-DMP; (4)–staining of proteins with Coomassie Brilliant Blue R-250.

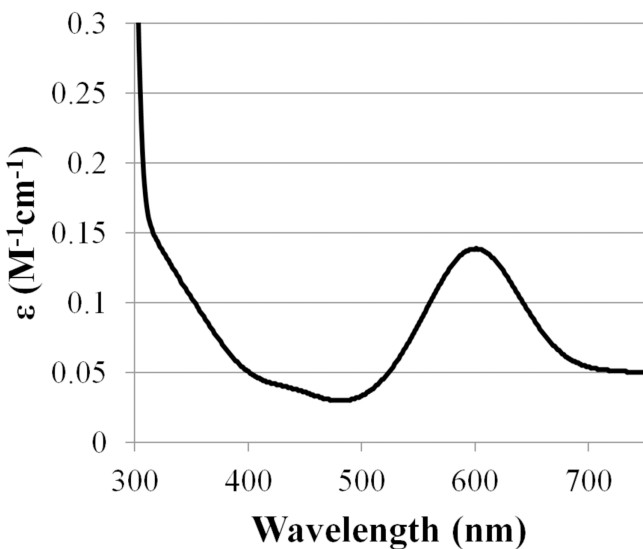

**Figure 2 Absorption spectrum of CjSL.**

phenanthroline acts as a good inhibitor in both the acidic and alkaline pH value (at 10 mM, the residual activity was 2.5% and 5.7% at pH 3.6 and 9.2, respectively). $NaN_3$, NaCl and NaF reduced laccase activity only at acidic conditions (Table 1), and $NaN_3$ was the most effective laccase inhibitor (at 10 mM, the residual activity of the laccase was less than 2%). At pH 9.2, $NaN_3$ decreased the laccase activity only at 100 mM, NaF did not display inhibitory effect, and NaCl had a low inhibitory effect (by 5% at 100 mM). EDTA

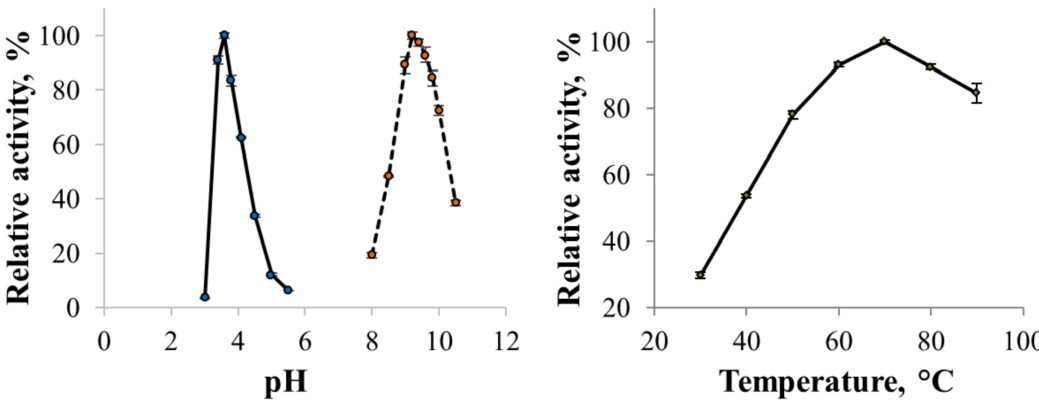

**Figure 3** **pH optima of the enzyme with ABTS and 2,6-DMP (A) and effect of temperature on laccase activity (B).** Error bars represent standard deviations for three replicates.

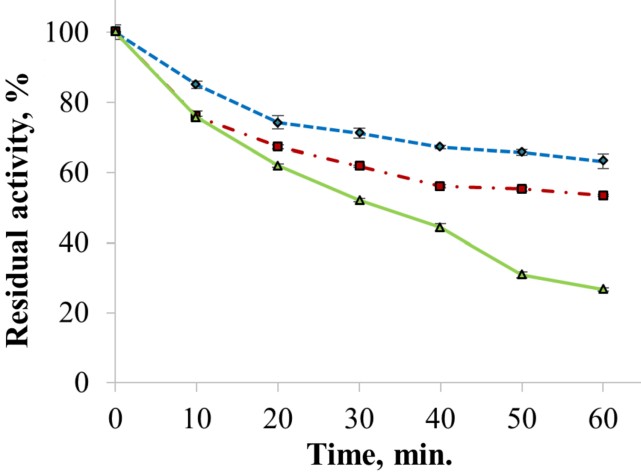

**Figure 4** **Thermal stability of laccase at 80 (diamonds), 90 (squares) and 100 °C (triangles).** Error bars represent standard deviations for three replicates.

dramatically reduced laccase activity at alkaline conditions (residual activity less than 9% at 10 mM). At pH 3.6, EDTA inhibited the laccase to a much lesser extent (50% of residual activity at 10 mM).

Investigation of the enzyme's substrate specificity showed that CjSL was able to oxidize 2,6-dimethoxyphenol, 3,4,5-trimethoxyphenol, 2-aminophenol, catechol, guaiacol, hydroquinone, pyrogallol, methylhydroquinone, syringaldehyde, syringaldazine, caffeic acid, o-coumaric acid, ferulic acid, gallic acid, gentisic acid, syringic acid, tannic acid, vanillic acid, 3,4-dihydroxybenzoicacid, ABTS and $K_4[Fe(CN)_6]$. The enzyme did not oxidize L-tyrosine, vanillin, 2-thiobarbituric acid, 3,5-dimethoxyphenol and 4-hydroxybenzoic acid. Table 2 summarizes the results.

The kinetic constants of CjSL were determined with ABTS (pH 3.6) and 2,6-DMP (pH 9.2), For ABTS, the $K_m$ was 0.39 mM; $V_{max}$ = 13.09 nmol/min. For 2,6-DMP, the $K_m$ was 1.86 mM; $V_{max}$ = 2.93 nmol/min.

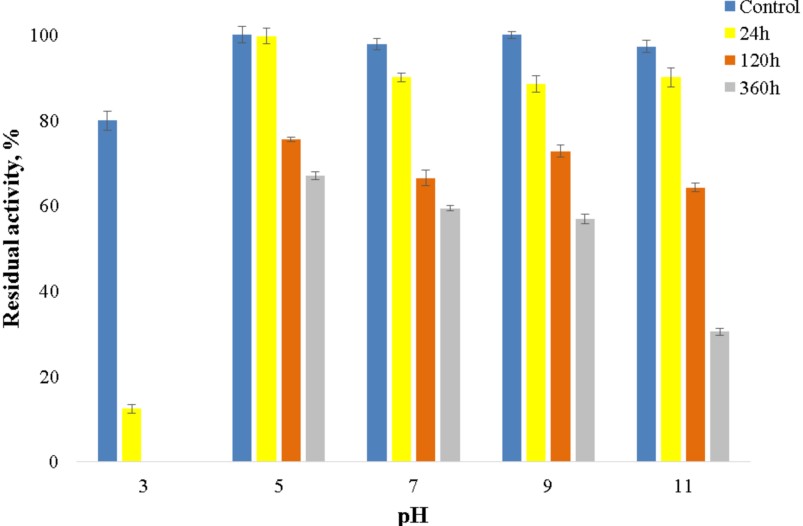

**Figure 5 pH stability of laccase.**               

**Table 1 The effect of some conventional inhibitors on laccase activity.**

| Compound | Concentration | Residual activity, % | |
|---|---|---|---|
| | | pH 3.6 | pH 9.2 |
| NaN$_3$ | 1 mM | 39 ± 0.8[a] | 102.4 ± 1.5 |
| | 10 mM | 1.2 ± 0.1 | 105.4 ± 1.9 |
| | 100 mM | 0 | 74.5 ± 1.3 |
| NaF | 1 mM | 77.7 ± 0.3 | 99.0 ± 0.5 |
| | 10 mM | 32.4 ± 0.1 | 100.7 ± 0.9 |
| | 100 mM | 0.9 ± 0.1 | 105.5 ± 0.8 |
| NaCl | 1 mM | 68.2 ± 0.9 | 97.4 ± 0.4 |
| | 10 mM | 26.3 ± 0.8 | 98.6 ± 0.1 |
| | 100 mM | 4 ± 0.7 | 95.6 ± 0.8 |
| EDTA | 1 mM | 100.5 ± 0.6 | 84.4 ± 0.2 |
| | 10 mM | 86.8 ± 0.6 | 8.8 ± 0.3 |
| | 100 mM | 50.8 ± 0.1 | 0 |
| 1,10-phenanthroline | 1 mM | 62.0 ± 0.6 | 84.6 ± 0.9 |
| | 10 mM | 2.5 ± 0.1 | 55.7 ± 0.4[b] |
| | 100 mM | 0 | |

**Note:**
[a] Here and below: standard deviations were calculated for at least three replicates.
[b] In this case, a precipitate was formed, which prevented spectrophotometric determination.

## Dye Decolorization

The purified laccase alone could efficiently decolorize only azo dye Congo Red at pH 4.0 (Table 3). Decolorization rate of Congo Red by laccase was 36%/day. Other dyes, namely Malachite Green, Brilliant Green, Methyl Orange, and Methyl Red, were unoxidizable by laccase without a mediator or oxidized with a low rate (4%/day and less) (Table 3). Laccase could decolorize triphenylmethane dyes, Malachite Green and Brilliant Green, only in presence of redox mediator ABTS. As shown in Table 3, 22% and 94%

**Table 2 Laccase activity against different types of substrates.**

| Substrate | $\lambda_{max}$, nm | Arb. units[*] | Substrate | $\lambda_{max}$, nm | Arb. units |
|---|---|---|---|---|---|
| 2,6-dimethoxyphenol | 469 | 3.044 ± 0.035 | Caffeic acid | 400 | 0.057 ± 0.001 |
| 3,5-dimethoxyphenol | − | − | o-coumaric acid | 360 | + |
| 3,4,5-trimethoxyphenol | 260 | 0.099 ± 0.002 | Ferulic acid | 420 | 0.248 ± 0.016 |
| 2-aminophenol | 420 | 2.622 ± 0.021 | Gallic acid | 350 | 0.031 ± 0.001 |
| Catechol | 252 | 0.056 ± 0.001 | Gentisic acid | 270 | 0.066 ± 0.002 |
| Guaiacol | 470 | 0.101 ± 0.001 | Syringic acid | 300 | + |
| Hydroquinone | 289 | 0.889 ± 0.050 | Tannic acid | 420 | 0.008 ± 0.001 |
| Pyrogallol | 335 | 0.570 ± 0.061 | vanillic acid | 340 | + |
| Methylhydroquinone | 287 | 1.252 ± 0.013 | 2-thiobarbituric acid | − | − |
| Syringaldehyde | 276 | 0.015 ± 0.001 | 3,4-dihydroxybenzoic acid | 350 | + |
| Syringaldazine | 330 | 0.261 ± 0.011 | 4-hydroxybenzoic acid | − | − |
| Vanillin | − | − | ABTS | 420 | 12.484 ± 0.116 |
| Tyrosine | − | − | $K_4[Fe(CN)_6]$ | 420 | 0.433 ± 0.049 |

Note:
[*]Laccase activity is expressed in arbitrary units (1 unit equals to change of absorbance by 0.1 optical absorption unit per minute); [+]changes were detected in the absorption spectra after 24 h; [−]no changes in the absorption spectra were detected.

**Table 3 Decolorization of dyes by CjSL without or with mediators, at different pH and 30 °C.**

| Decolorization of dye, % | | | | | |
|---|---|---|---|---|---|
| | Congo red | Methyl orange | Methyl red | Brilliant green | Malachite green |
| pH 4.0 | | | | | |
| CjSL | 36 ± 4 | 5 ± 1 | 3[a] | 0 | 0 |
| CjSL + ABTS | 0 | 57 ± 2 | 12 | 94 | 22 |
| CjSL + SA | 29 ± 6 | 6 ± 1 | 8 | 4 | 0 |
| pH 6.5 | | | | | |
| CjSL | 4 ± 1 | 2 | 3 ± 1 | b | b |
| CjSL + ABTS | 31 | 7 | 46 | b | b |
| CjSL + SA | 22 | 3 | 15 ± 4 | b | b |
| pH 9.2 | | | | | |
| CjSL | 2 | 2 | 4 ± 2 | b | b |
| CjSL + ABTS | 8 | 3 | 10 | b | b |
| CjSL + SA | 8 | 2 | 10 ± 1 | b | b |

Note:
[a]Standard deviation below 1%; [b]No measurements were carried out due to non-enzymatic oxidation of the dyes at these conditions.

decolorization for Malachite Green and Brilliant Green (50 μM), respectively, was obtained within 24 h incubation with 200 mU ml$^{-1}$ of laccase in 50 mM Britton and Robinson buffer at 30 °C. Azo dye Methyl Orange was bleached with the highest rate at pH 4.0 in presence of ABTS (decolorization rate was 57%/day), whereas in presence of syringaldehyde the rate was quite low (6%) (Table 3). The highest decolorization rate for azo dye Methyl Red was observed at pH 6.5 in presence of ABTS and comprised 46%/day.

At pH 4.0 and 9.2, the rate of Methyl Red decolorization in presence of ABTS was lower, comprising 12%/day and 10%/day, respectively. Congo Red with syringaldehyde and ABTS as redox mediators was most intensely decolorized at pH 6.5 (decolorization rate comprised 31% and 22%, respectively).

## DISCUSSION

In this study, we cloned a new laccase gene from *Catenuloplanes japonicus* VKM Ac-875. Sequence analysis showed that it belongs to B-type two-domain laccases according to classification by *Nakamura & Go (2005)*: it possesses binding sites for T1-copper center only in the second domain. In the structural aspect, CjSL comprised a classical two-domain enzyme having conserved copper-binding amino acids (ten histidines and one cysteine), and a signal peptide (first 45 amino acid residues) facilitating transmembrane TAT-dependent translocation as well. Since the expression of two-domain laccase with signal peptide sequence can lead to absence of enzyme production, as it was shown for Ssl1 (*Gunne & Urlacher, 2012*), *cjsl* gene was cloned into expression vector without the sequence of signal peptide. For expression of two-domain laccase, pQE-30 vector was chosen, and the laccase gene was cloned under T5 promoter control. A conventional scheme for production of two-domain laccases was used, which resulted earlier in high yield of the enzymes (up to 180 mg per liter of culture) (*Trubitsina, Trubitsin & Lisov, 2020*). However, the expression of CjSL had very low yield, around 5 mg laccase per liter. Modification of the initial technique, i.e., growth of culture up to $OD_{600} = 0.2–0.25$, and elevation of aerobic induction temperature up to 22 °C, resulted in increase of enzyme yield up to 10 mg per liter. It is known that oxidation of phenolic compounds serving as both electron and proton donors by three-domain fungal laccases is pH-dependent, and the dependency is a bell-like curve. This is brought forward by, on one hand, the fact that ionization potential of the substrate is decreasing with increasing pH because of formation of phenolate ions, which leads to increase of oxidation rate of the latters. On the other hand, inhibition of laccase by $OH^-$-anions at elevated pH is caused by their strong binding to copper atoms in T2/T3-cluster of the enzyme. Thus, the bell-like shape of the oxidation curve of phenolic compounds by laccases is caused by combined action of two opposite effects (*Xu, 1997*). In the case of two-domain laccases, including the described CjSL enzyme, a similar dependence takes place. But when considering the studied non-phenolic compounds (donors of electrons but not protons), pH change of the medium does not alter their ionization potential, and the dependency of oxidation of this type of enzymes on pH of the medium would be determined by only one effect, inhibition by $OH^-$-anions, and thus the dependency curve should be monotonically descending. Such a dependency is specific for three-domain fungal laccases. However, in case of two-domain bacterial laccases, the non-phenolic compound oxidation curve is also bell-like. The reason for this could be low stability of the enzymes at acidic pH values.

Table 4 demonstrates comparative characterization of CjSL and two-domain laccases studied earlier. The ability of CjSL to retain the activity even at boiling represents the enzyme as a very thermostable protein. The novel enzyme is the most thermostable two-domain laccase studied up to date. Contrary to the majority of two-domain laccases,

**Table 4 Comparative characterization of two-domain bacterial laccases (subgroup B).**

| Enzyme, bacteria (reference) | Molecular weight | pH optimum | pH stability | Thermal stability/ optimum T | Kinetic parameters |
|---|---|---|---|---|---|
| CjSL, *Catenuloplanes japonicas* (this work) | 37 kDa (SDS-PAGE)[a] 200 kDa (native-PAGE) | ABTS–3.6; 2,6-DMP–9.2 | 5–7 | $t_{1/2}$ at boiling for 30 min $t_{1/2}$ at 90 °C for 1h /70 °C | ABTS $K_m$ 0.39 mM, $k_{cat}$ 6.83 $s^{-1}$ 2,6-DMP: $K_m$ 1.86 mM, $k_{cat}$ 0.95 $s^{-1}$ |
| SpSL, *Streptomyces puniceus* (*Trubitsina et al., 2020*) | 40 kDa (SDS-PAGE) 110 kDa (gel-filtration) | ABTS–3.5; 2,6-DMP–9.0 | 6–9 | $t_{1/2}$ at 80 °C for 40 min $t_{1/2}$ at 90 °C for 20 min /90 °C | ABTS: $K_m$ 0.37 mM, $k_{cat}$ 24.3 $s^{-1}$ 2,6-DMP: $K_m$ 1.15 mM, $k_{cat}$, 3.4 $s^{-1}$ |
| SaSL, *S. anulatus* (*Lisov et al., 2019*) | 40 kDa (SDS-PAGE) 235 kDa (gel-filtration) | ABTS–3.0; 2,6-DMP–8.5 | 8–11 | $t_{1/2}$ at 80 °C for 40 min $t_{1/2}$ at 90 °C for 20 min /85 °C | ABTS: $K_m$ 0.17 mM, $k_{cat}$ 6.3 $s^{-1}$ 2,6-DMP: $K_m$ 1.75 mM, $k_{cat}$, 1.42 $s^{-1}$ |
| MCO, *S. griseorubens* (*Feng et al., 2015*) | 33.9 (SDS-PAGE) | ABTS–4.0; 2,6-DMP–9.0 | 7–11 | $t_{1/2}$ at 40 °C for 10 h $t_{1/2}$ at 70 °C for 2 h/[c] | ABTS: $K_m$ 22.3 $s^{-1}$ $mM^{-1}$, $k_{cat}$ 7.68 $s^{-1}$ 2,6-DMP: $K_m$ 0.39 $s^{-1}$ $mM^{-1}$, $k_{cat}$, 0.33 $s^{-1}$ |
| SvSL, *S. viridochromogenes* (*Trubitsina et al., 2015*) | 34 kDa (SDS-PAGE); 99 kDa (PDB: 4N8U) | ABTS–4.0; 2,6-DMP–8.5 | 6–10 | $t_{1/2}$ at 80 °C for 30 min $t_{1/2}$ at 90 °C for 20 min /90 °C | ABTS: $K_m$ 0.3 mM, $k_{cat}$ 8 $s^{-1}$ 2,6-DMP: $K_m$ 4.5 mM, $k_{cat}$, 1.9 $s^{-1}$ |
| LMCO, *S. pristinaespiralis* (*Reiss et al., 2013*; *Ihssen et al., 2015*) | 38 kDa (SDS-PAGE) | ABTS–4.7; 2,6-DMP–7.6 | [c] | $t_{1/2}$ at 70 °C for 30 min/[c] | [c] |
| Ssl1, *S. sviceus* (*Gunne & Urlacher, 2012*) | 33 kDa (SDS-PAGE); 99 kDa (PDB: 4M3H) | ABTS–4.0; 2,6-DMP–9.0 | 11 | $t_{1/2}$ at 70 °C for 29 min $t_{1/2}$ at 80 °C for 10 min/[c] | ABTS: $K_m$ 0.36 mM, $k_{cat}$ 7.38 $s^{-1}$ 2,6-DMP: $K_m$ 0.89 mM, $k_{cat}$, 0.32 $s^{-1}$ |
| SilA, *S. ipomoea* (*Molina-Guijarro et al., 2009*) | 44.7 kDa (SDS-PAGE) | ABTS–5.0; 2,6-DMP–8.0 | 5–9 | $t_{1/2}$ at 60 °C for 24 h /60 °C | ABTS: $K_m$ 0.4 mM, $k_{cat}$ 9.99 $s^{-1}$ 2,6-DMP: $K_m$ 4.27 mM, $k_{cat}$, 4.2 $s^{-1}$ |
| SLAC *S. coelicolor* (*Dube et al., 2008*) | 32 kDa (SDS-PAGE) 99 kDa (PDB: 3CG8) | ABTS–4.0; 2,6-DMP–9.0 | 3–9 | $t_{1/2}$ at 70 °C for 110 min /60 °C | [c] |
| EpoA, *S. griseus* (*Endo et al., 2003*) | 38 kDa (SDS-PAGE) 114 kDa (native SDS-PAGE) | DMP[b]–6.5 | [c] | $t_{1/2}$ at 70 °C for 40 min /40 °C | DMP: $K_m$ 0.42 mM, $V_{max}$ 0.85 nM/min |

**Note:**
[a]The method for determining the molecular weight of a protein is indicated in parentheses; [b]N,N-dimethyl-p-phenylenediamine sulfate; [c]No data available.

which are acting as trimers, CjSL is active in the state of higher oligomerization degree. Its zymogram indicates the presence of one colored band of approximate size of 200 kDa. Similar molecular properties were revealed earlier only in one two-domain laccase, SaSL from *Streptomyces anulatus* Ac-728 VKM (*Lisov et al., 2019*). As it is shown in Table 4, the majority of two-domain laccases is more stable at alkaline pH values. At pH 5.0 and lower, the enzymes are rapidly inactivated. CjSL laccase is maximally active at pH 5.0. High stability at pH 5.0 and below was earlier shown only for two-domain laccase SilA (*Molina-Guijarro et al., 2009*) and SLAC (*Dube et al., 2008*).

Inorganic ions like fluoride-, chloride-, azide-can bind to T2/T3 active center of laccase and block the electron transfer from T1 copper center (*Xu, 1996*; *Gianfreda, Xu & Bollag, 1999*). Most of the three-domain fungal laccases are completely inhibited by 1 mM sodium azide (*Liu, Deng & Yang, 2021*; *Nitheranont, Watanabe & Asada, 2011*).

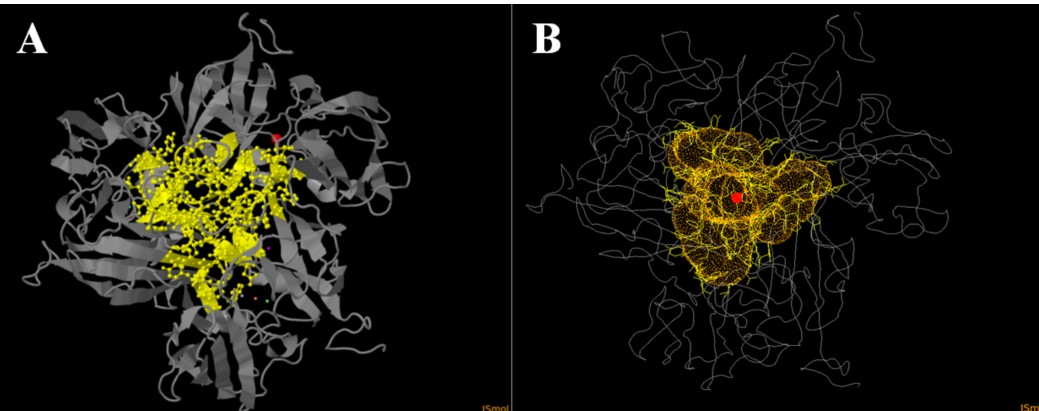

**Figure 6 Laccase tunnel mapping.** (A) Tunnel-lining residues (yellow balls and sticks) and copper atoms (violet dot–T1 copper; blue, pink and green dots–T2/T3 copper atoms). The red ball shows the origin (mouth) of the tunnel. (B) Tunnel in Ac-875 laccase model (homotrimeric complex). The red circle shows the origin (mouth) of the tunnel. The orange indicates the surface of the tunnel. Lining residues are shown as yellow lines.

But this inhibition effect of azide ions was not observed in case of CjSL (at alkaline pH). Absence of inhibiting effect or weak inhibiting effect of azide, chloride or fluoride on the activity of two-domain laccases at alkaline pH was earlier shown by other research groups. For example, SilA was not inhibited by chloride ions at pH 8.0 even at 1 M NaCl concentration. At pH 5.0, the residual activity in presence of 1 M NaCl was 64.4% (*Molina-Guijarro et al., 2009*). In presence of 10 mM $NaN_3$, Ssl1 retained 60.3% and 94.9% of the initial activity at pH 5.0 and 8.0, respectively. Ssl1 preserved 95% of its initial activity at 10 mM $NaN_3$ at pH 9.0 (*Gunne & Urlacher, 2012*). MCO laccase from *Streptomyces griseorubens* retained 94.5% of the initial activity at pH 9.0 in presence of 1 mM sodium azide (*Feng et al., 2015*). We also established earlier, that SvSL was not inhibited by fluoride-and azide-ions at alkaline conditions. Inhibition of the laccase activity at pH 4.0 was weak with $IC_{50}$ of 7 mM for azide and 15mM for fluoride (*Trubitsina et al., 2015*). EDTA and 1,10-phenanthroline, which act as chelating agents, decreased laccase activity via binding active site copper ions. The inhibiting effect of phenanthroline did not depend on pH. It could be related to high affinity of phenanthroline to copper ions even at pH 3.6, when around 20% of phenanthroline molecules are charged (the pKa of phenanthroline is 4.2). In case of EDTA, the inhibitor acted stronger at alkaline pH values. The analogous results were shown for SilA (EDTA decreased its activity significantly only at alkaline pH) (*Molina-Guijarro et al., 2009*). This can be explained by the ability of EDTA to bind metals most effectively in its four-carboxylate form, and its pKa equals to 10.26.

    Since inhibition by sodium fluoride and azide was pH-dependent, we thought that it is related to differences in diffusion properties of fluoride and azide at basic and acidic pH values. Since pKa of hydrazoic acid equals to 4.6 and pKa of hydrofluoric acid equals to 3.17. Our initial hypothesis was that an inhibitor acts only in protonated form when diffuses into the channel and is deprotonated subsequently right before attack onto Cu atoms. And indeed, we found that tunnel mapping in CAVER online showed a specific channel in one model. It had trilateral symmetry (Fig. 6), meaning that the lining residues

**Table 5 Types, numbers and positions of charged residues surrounding the channel in Ac-875 laccase.**

| Type of charged groups | Positive amino acids (K, R only) | Positive amino acids (K, R plus H) | Negative amino acids (D, E only) | Negative amino acids (D, E plus Y) |
|---|---|---|---|---|
| Number of charged residues | 2 | 4 | 5 | 7 |
| Positions of charged residues | R199, K216 | H59, H109, R199, K216 | D197, E204, D208, D214, D217 | Y107, Y185, D197, E204, D208, D214, D217 |
| pKa of charged groups[a] | pKa ( K ) = 10.53 pKa ( R ) = 12.48 | pKa ( K ) = 10.53 pKa ( R ) = 12.48 pKa ( H ) = 6.00 | pKa ( D ) = 3.65 pKa ( E ) = 4.25 | pKa ( D ) = 3.65 pKa ( E ) = 4.25 pKa (Y) = 10.5[b] |

Note:
[a] according to http://www.chem.ucalgary.ca/courses/351/Carey5th/Ch27/ch27-1-4-2.html.
[b] according to http://www.chem.ucla.edu/~rebecca/153A/amino_acids.pdf.

were identical in all the three subunits of the enzyme, thus it seems to be relevant enough to the actual trimeric complex. The lining residues of the tunnel are as follows: H (59) GYWH (106–109) Y (185) WADNR (195–199) E (204) TDP (207–209) ASIDN (211–215) KDLG (216–219) GSSFG (221–225).

The numbers and positions of charged residues are listed in Table 5. As we can conclude from Table 5, the overall charge of the channel can be neutral below pH 4, and hydrogen fluoride and hydrazoic acid are also present in neutral forms at these pH values in reasonable amounts (around 80% hydrazoic acid and 16% hydrogen fluoride are neutral). So, the initial hypothesis got its proofs at the first stages of our experiments. However, it was surprising to see that sodium chloride also displays laccase-inhibiting properties at acidic pH, and it is almost as effective as fluoride in laccase inhibition. Since hydrochloric acid is strong, the hypothesis on mandatory diffusion of the inhibitor in protonated form is not supported by the experiments with sodium chloride. Hence, we must suggest that protonation and neutralization of tunnel lining residues should be enough to facilitate the diffusion of azide, fluoride and chloride to the copper atoms of the active site of Ac-875 laccase.

Fungal laccases are known to have broad substrate specificity due to their high redox potential (*Mot & Silaghi-Dumitrescu, 2012*). Bacterial laccases have low redox potential (Durao et al., 2006; Gunne et al., 2014; *Blánquez et al., 2019*), so they can oxidize a narrower range of organic and inorganic compounds. CjSL laccase has been shown to be able to oxidize model laccase substrates: ABTS, 2,6-DMP, guaiacol and syringaldazine. The enzyme oxidized wide range of aromatic carboxylic acids (caffeic, o-coumaric, ferulic, gallic, gentisic, 3,4-dihydroxybenzoic, syringic and vanillic acids), aromatic azo compounds (ABTS and syringaldazine), aromatic alcohols (catechol, guaiacol, hydroquinone, pyrogallol, 2,6-dimethoxyphenol, methylhydroquinone, 2-aminophenol, 3,4,5-trimethoxyphenol), aromatic aldehyde (syringaldehyde), polyphenol compound tannic acid and inorganic compound $K_4[Fe(CN)_6]$. Oxidation of aromatic compounds was observed only at alkaline pH values. At acidic pH, these compounds were not oxidized by the laccase (the activity was nil). An attempt to measure the laccase redox potential by redox titration with $K_3[Fe(CN)_6]/K_4[Fe(CN)_6]$ redox pair gave no results. Thus, the redox

potential of CjSL laccase is either below 0.31 mV or above 0.46 mV. However, considering the broad substrate specificity of the enzyme, we suggest that its redox potential is higher than that of other two-domain laccases (>0.46 mV).

To study the ability of the two-domain laccase to degrade stable compounds, we chose the following dyes: Congo Red, a diazo dye used in textile industry, which has carcinogenic and cytotoxic properties (*Hernandez-Zamora et al., 2016*); Methyl Orange, a carcinogenic sulfonated mono azo dye (*Haque et al., 2021*); Methyl Red, anionic azo dye causing irritation of eyes, skin, and gastrointestinal tract (*Badr, Abd El-Wahed & Mahmoud, 2008*); Malachite Green, a triphenylmethane dye with toxic effects on various fish species and certain mammals (*Srivastava, Sinha & Roy, 2004*); Brilliant Green, another triphenylmethane dye. It was established that two-domain laccase CjSL is able to decolorize azo dyes alone with low efficacy (5% and less) (Table 3). The exception is Congo Red decolorization at pH 4.0 (36 ± 4%/day). pH value affected decolorization rate. So, Methyl Orange was bleached more effectively at pH 4.0, whereas Methyl Red was better oxidized at pH 6.5. Both mediators, ABTS and SA, elevated dye decolorization rate of laccase, but ABTS was more effective mediator. These results point at the possibility of laccase application for dye decolorization; further studies are required to optimize this process and enhance its efficacy. For this purpose, a wider range of dyes and natural and synthetic mediators is planned to be used at different concentrations, and the dye degradation reactions should be performed at different temperature and pH values.

## CONCLUSIONS

CjSL is a newly characterized two-domain laccase from actinobacteria *Catenuloplanes japonicus*. High thermostability, pH-stability, ability to oxidize a wide range of different phenolic substrates and resistance to inorganic ions (fluoride-, azide-, chloride-) at alkaline conditions are the most important features of this enzyme. CjSL laccase is able to decolorize triphenylmethane dyes in couple with synthetic redox mediator ABTS and azo dyes in couple with ABTS or natural redox mediator, syringaldehyde. Thus, the novel two-domain laccase can be regarded as a promising enzyme for biotechnological applications.

### Funding

This work was supported by the Ministry of Science and Higher Education of the Russian Federation as part of a state assignment in the field of scientific activity (Project No. FEWG-2020-0008). The funders had no role in study design, data collection and analysis, decision to publish, or preparation of the manuscript.

### Grant Disclosures

The following grant information was disclosed by the authors:
Ministry of Science and Higher Education of the Russian Federation: FEWG-2020-0008.
## Competing Interests

The authors declare that they have no competing interests.

## Author Contributions

- Liubov Igorevna Trubitsina conceived and designed the experiments, performed the experiments, analyzed the data, prepared figures and/or tables, authored or reviewed drafts of the paper, and approved the final draft.
- Azat Vadimovich Abdullatypov conceived and designed the experiments, analyzed the data, prepared figures and/or tables, authored or reviewed drafts of the paper, and approved the final draft.
- Anna Petrovna Larionova performed the experiments, prepared figures and/or tables, and approved the final draft.
- Ivan Vasilyevich Trubitsin performed the experiments, prepared figures and/or tables, and approved the final draft.
- Sergey Valerievich Alferov analyzed the data, authored or reviewed drafts of the paper, and approved the final draft.
- Olga Nikolaevna Ponamoreva analyzed the data, authored or reviewed drafts of the paper, and approved the final draft.
- Alexey Arkadyevich Leontievsky conceived and designed the experiments, authored or reviewed drafts of the paper, and approved the final draft.

## Data Availability

The raw measurements, electrophoregrams and a zymogram, and a model of Ac-875 laccase used in tunnel mapping are available in the Supplemental Files.

## Supplemental Information

Supplemental information for this article can be found online at http://dx.doi.org/10.7717/peerj.11646#supplemental-information.

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
