# Peer review of "Expression of thermophilic two-domain laccase from Catenuloplanes japonicus in Escherichia coli and its activity against triarylmethane and azo dyes"

_PeerJ, doi:10.7717/peerj.11646_

## Round 0.1 · original submission · Minor Revisions

Please, find below the comments from the Reviewers.

Reviewer 1 ·

Basic reporting

This paper describes a newly characterized two-domain laccase from аctinobacteria Catenuloplanes japonicas. This laccase is able to decolorize triarylmethane and azo dyes and could be considered as a promising enzyme for biotechnological applications because of its high thermostability, рН-stability, ability to oxidize different phenolic substrates and resistance to inorganic ions at alkaline conditions.

In general, the information enclosed in this work is original and valuable.
The introduction and the background are all right and show context. The literature is well referenced and relevant. The manuscript structure follows to PeerJ standards, and the raw data supplied is satisfactory.

Nevertheless the English language should be improved. Some examples where the language could be improved include lines 57-59 and 86-90, the current phrasing makes comprehension difficult.
Figures are relevant, high quality, and generally well labeled.

Experimental design

The primary research is within the Scope of the journal.
The Research question is well defined and relevant.

The methods are described with sufficient detail and information to replicate.

Validity of the findings

The impact of the findings is assessed.

All underlying data have been provided and they are robust. However statistical analysis of data is not mentioned in material and methods, neither in results. The number of assay replicates should be mentioned and standard deviation added.

Conclusions are well stated and linked to original research question.

Additional comments

Line 27: “inhibitors of classical fungal laccases (azides, fluorides), Low redox potential was found…
Please replace the comma before “Low” with a period.

Line 38: “ A novel two-domain recombinant laccase CjSL turned out…”
The term “turned out” in this phrase is inadequate, please write it in another way.

Line 43-44: “The pH optima of CjSL for oxidation of inorganic and organic substrates were 3.6 and 9.2 respectively”
Please replace with: “The pH optima of CjSL for oxidation of ABTS and 2,6-DMP substrates were 3.6 and 9.2 respectively”

Lines 56-60
“It contains four copper ions in the active site. Four copper atoms arrange into three types of metal centers (T1, T2 and T3) take part in catalytic act (Solomon et al., 1996; Bento et al., 2005). Meanwhile, four electrons which are necessary for complete reduction of oxygen to water laccase are subsequently taken away from the substrates (a wide range of organic and inorganic compounds) (Morozova et al., 2007).”
This sentence is not clear, please rephrased it.

Line 69 Only bacterial laccases “can” contain two or three structural and functional domains.
Please delete “can”.

Line 78 It was used (in complex with mediator) in detoxification of dyes.
Please replace with: They are used (in complex with mediator) in detoxification of dyes.

Line 83 members of this group, searching, obtaining and investigation of new two-domain
proteins…
Please replace with: members of this group, searching, obtaining and “investigating” of new two-domain proteins…

Line 94 “Microorganism, cloning of cjsl gene”
Modify to: Microorganism, “and” cloning of cjsl gene

Line 112 “plasmid pQE/cjsl”
Please replace plasmid pQE/cjsl with pQE::cjsl

Line 120
Why do you use such long induction times?

Line 162 “putative substrates by detecting the changes in the absorption spectrum after a 24-h incubation”.
Please replace 24-h with 24 h

Line 163 Compounds were assayed at pH 3.6 or pH 9.2 for non-phenolic and phenolic compounds…
Please replace with: “The substrate specificity was” assayed at pH 3.6 or pH 9.2 for non-phenolic and phenolic compounds…

In Parragraph 162-173 (and all over the ms)
Please replace period with comma in substrate names, for example: 3.4.5-trimethoxyphenol with 3,4,5-trimethoxyphenol

Line 173 The substrate concentration of 1 mM was used.
Please replace with: The substrate concentration used was 1 mM.

Line 203 An experimentally shown structure of another trimeric…
Please replace with: An experimentally determined structure of another trimeric …

Line 257 At pH 9.2, “NaN3 decreased” the laccase

Line 275 The purified laccase could efficiently decolorize only azo dye Congo Red at pH 4.0 when
taken alone.
Please replace with: The purified laccase “alone” could efficiently decolorize only azo dye Congo Red at pH 4.0.

Line 293 In this study, we cloned “a” new laccase gene from Catenuloplanes japonicas…

Line 296 … copper center only in the second domain. In “the” structural aspect,


Line 301 … elevation of aerobic induction temperature up to 22°C, resulted in increase...
Induction conditions are not coincident between this line and lines 120-121. Was the induction temperature 22°C or 18°C and 25°C? Please clarify the situation.

Lines 351-355 The grammar of this sentence should be improved.

Line 379 Fungal “laccases are” known to

Table 2:
1 unite quals = 1 unit equals??

Table 4:
Where no data available, replace – by c.

Table 5:
Replace: Positive amino acids (with histidine) with Positive amino acids (K,R with/plus H)

Replace: Negative amino acids (with tyrosine) with Negative amino acids (D, E with/plus Y)

Replace Pka with pKa

Table 1-4 The reported values are the mean of how many replicates? Add this info to mat and methods, and in table legend. Please add standard deviations.

Figure 1:
Replace 2.6-DMP with 2,6-DMP

Figure 2:
Replace comma with dots

Figure 3A Lines labels are missing. Please explain which line corresponds to each substrate.


Figure 4 and 5: How many replicates did you do? Please add error bars.

Figure 5: Please explain what K means. Control?


I believe that with this corrections, and the improvement of the English language and grammar the manuscript will increase its quality and would be suitable for publication in PeerJ.

·

Basic reporting

without comments

Experimental design

without comments

Validity of the findings

without comments

Additional comments

The work is a very good contribution regarding the type of laccase obtained for various biotechnological applications. The authors cloned the laccase from Catenuloplanes japonicus in Coli.
The laccase obtained has high thermostability, рН-stability and the ability to oxidize a wide range of different phenolic substrates and resistance to inorganic ions at alkaline conditions. Also, is able to decolorize some dyes. The manuscript is well organized and structured. The bibliography is current. The methodology is very well described and complete.
The results are thoroughly analyzed and integrated. I have no comments about the manuscript.

---

## Round 0.2 · accepted · Accept

The authors have made the corrections suggested by the reviewers. The current version of the manuscript has improved.